# Pedestrian Detection Based on Two-Stream UDN

**Wentong Wang [1,2,3]**, **Lichun Wang [1,2,3]**, **Xufei Ge [4]**, **Jinghua Li [1,2,3],\*** and **Baocai Yin [1,2,3]**

[1]  Beijing Key Laboratory of Multimedia and Intelligent Software Technology, Beijing University of Technology, Beijing 100124, China; wwt.phd@emails.bjut.edu.cn (W.W.); wanglc@bjut.edu.cn (L.W.); ybc@bjut.edu.cn (B.Y.)
[2]  Beijing Artificial Intelligence Institute, Beijing University of Technology, Beijing 100124, China
[3]  Faculty of Information Technology, Beijing University of Technology, Beijing 100124, China
[4]  CITIC Guoan Broadcom Network Co., Ltd., Beijing 100176, China; gexufei@citicguoanbn.com
\*  Correspondence: lijinghua@bjut.edu.cn



**Featured Application: This paper can be applied to an autonomous vehicle and driving assistance system.**

**Abstract:** Pedestrian detection is the core of the driver assistance system, which collects the road conditions through the radars or cameras on the vehicle, judges whether there is a pedestrian in front of the vehicle, supports decisions such as raising the alarm, automatically slowing down, or emergency stopping to keep pedestrians safe, and improves the security when the vehicle is moving. Suffering from weather, lighting, clothing, large pose variations, and occlusion, the current pedestrian detection still has a certain distance from the practical applications. In recent years, deep networks have shown excellent performance for image detection, recognition, and classification. Some researchers employed deep network for pedestrian detection and achieve great progress, but deep networks need huge computational resources, which make it difficult to put into practical applications. In real scenarios of autonomous vehicles, the computation ability is limited. Thus, the shallow networks such as UDN (Unified Deep Networks) is a better choice, since it performs well while consuming less computation resources. Based on UDN, this paper proposes a new deep network model named two-stream UDN, which augments another branch for solving traditional UDN's indistinction of the difference between trees/telegraph poles and pedestrians. The new branch accepts the upper third part of the pedestrian image as input, and the partial image has less deformation, stable features, and more distinguished characters from other objects. For the proposed two-stream UDN, multi-input features including the HOG (Histogram of Oriented Gradients) feature, Sobel feature, color feature, and foreground regions extracted by GrabCut segmentation algorithms are fed. Compared with the original input of UDN, the multi-input features are more conducive for pedestrian detection, since the fused HOG features and significant objects are more significant for pedestrian detection. Two-stream UDN is trained through two steps. First, the two sub-networks are trained until converge; then, we fuse results of the two subnets as the final result and feed it back to the two subnets to fine tune network parameters synchronously. To improve the performance, Swish is adopted as the activation function to obtain a faster training speed, and positive samples are mirrored and rotated with small angles to make the positive and negative samples more balanced.

**Keywords:** pedestrian detection; Unified Deep Net; two-stream nets; network training

## 1. Introduction

Pedestrian detection is an important research field in computer vision. In recent years, it has become more and more widely used in vehicle-assisted driving, intelligent video surveillance, and

human behavior analysis. According to the Global Status Report on Road Safety 2018 published by the World Health Organization [1], 1.3 million people die every year in traffic accidents, and the distributions of traffic deaths of different types of road users in World Health Organization (WHO) regions are shown in Figure 1. Worldwide, 23% of the road traffic fatalities are pedestrians.

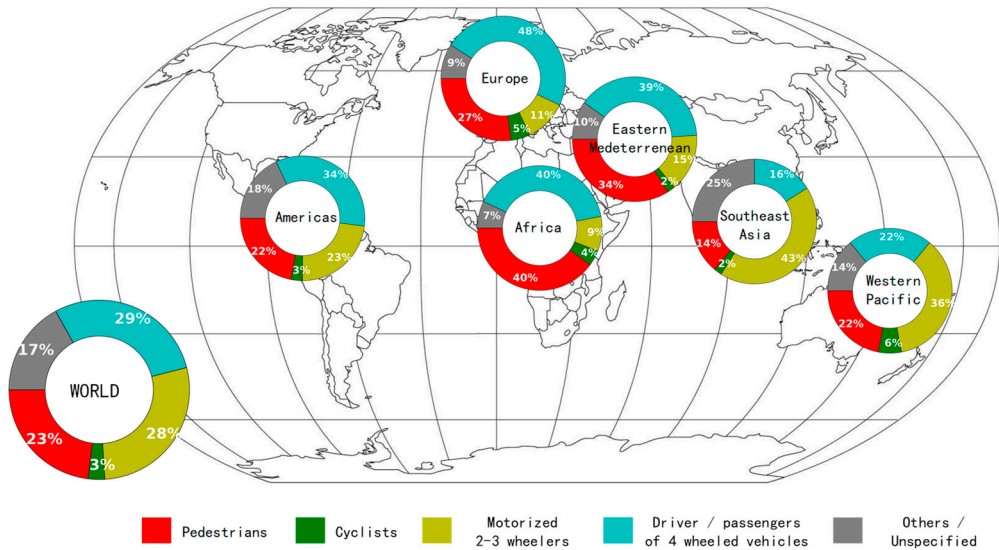

**Figure 1.** Distribution of deaths of road users in 2018 reported by the World Health Organization (WHO).

In order to reduce the occurrence of traffic accidents and protect the safety of pedestrians, major research institutions have conducted research on intelligent vehicle-assisted driving systems. The system includes a pedestrian detection system, which uses an on-board camera to obtain road condition information, applies an intelligent detection algorithm to detect pedestrians in the driving environment, reminds the driver to keep a safe speed, and promptly reports whether a collision may occur or performs deceleration and automatic braking.

As a hot issue in the field of computer vision, pedestrian detection attracted the attention of scholars in the mid-1990s. Difficulties in pedestrian detection includes the following: complex background in reality, the influence of weather or light, diversity of pedestrian poses, diversity of pedestrian clothing, occlusion of pedestrian and other objects, and different camera viewpoints. In order to solve the above problems, researchers have proposed a large number of algorithms to promote the effectiveness of pedestrian detection.

## 2. Related Works

The key to pedestrian detection is extracting pedestrian features, but a pedestrian is a non-rigid object that is complicated to represent. According to different ways of feature extraction, pedestrian detection methods can be divided into three types: template matching, statistical learning, and deep nets.

### 2.1. Template Matching-Based Pedestrian Detection

The pedestrian detection algorithm based on template matching aimed to establish a pedestrian target template database for different types of pedestrians. The template can be a pedestrian outline or a grayscale image. When detecting an image, the template features of the input image are calculated, and then searched for in a pedestrian template database to find whether a corresponding pedestrian template matches the input template feature successfully.

The contour-based hierarchical matching algorithm proposed by Gavrila [2,3] is a typical template-matching algorithm. The algorithm firstly used a layered matching method to lock the candidate area by using contour features of the human body, compute the distance transformation

image of the candidate area, and calculate the Chamfer distance between the distance transformation images and pedestrian templates. The second step used radial basis functions (RBF) with Chamfer distance to verify whether the candidate area is pedestrian. The candidate area includes persons, so the algorithm belongs to the overall template matching method.

In order to tackle occlusion and a variety of human poses, Shashua [4] proposes an algorithm based on human part template matching. The algorithm divided the human body into nine parts with overlapping regions and built nine types of pedestrian templates. Wu and Nevatia [5] divided the human body into three parts: a head with shoulders, a torso, and legs, and used Edgelet features for part detection. Each Edgelet is composed of a set of edge points to describe an outline of part of the human body.

Pedestrian detection based on a template-matching algorithm is relatively simple to calculate, but the algorithm requires a pre-designed pedestrian template database. The human body is a non-rigid object, and the pose changes are more complicated. Therefore, the algorithm has certain limitations. At the same time, the template-matching algorithms only consider the outline information of human body and ignore details such as the skin color and clothing of the human body. When the contour feature of pedestrians is relatively fuzzy, the performance would drop.

## 2.2. Statistical Learning-Based Pedestrian Detection

Methods based on statistical learning refer to learning a classifier through a series of training data and classified input regions with the learned classifier. Feature extraction and classifier design are core techniques of pedestrian detection based on statistical learning.

In the feature extraction stage, the key issue is extracting discriminative pedestrian features. A good feature should not only capture information different from other classes, but also maintain the stability of differences within class. Current feature acquisition methods for pedestrian detection include hand-crafted features and learned features. Hand-crafted features commonly include Haar-like features [6], SIFT (Scale-Invariant Feature Transform) features [7], HOG (Histogram of Oriented Gradients) features [8], and variations or combinations of these features. Using HOG is an important milestone in pedestrian detection, which highly promotes the pedestrian detection effect. CSS (color self-similarity) characterizing interrelationships between local block features [9] is combined with HOG to improve the detection performance greatly. Integral Channel Features [10] also achieved good detection results in pedestrian detection. Felzenszwalb et al. DPM (Deformable Part Model) [11,12], as a spring deformation model [13], is effective for solving pose changing in pedestrian detection.

In the classification stage, commonly used classifiers include SVM (Support Vector Machine) [14], Random Forest, Probabilistic Model, and Neural Network [15–17]. A combination of HOG and SVM is a classic algorithm in the history of pedestrian detection.

Benenson et al. [18] suggested that the classifier has less of an impact on pedestrian detection, and extracted features are more important. Pedestrian detection based on statistical learning adapts to images with simple backgrounds and less occlusion, but its effects need to be improved for images with complex backgrounds and large pose changing [15], so more robust features must be found.

## 2.3. Deep Learning-Based Pedestrian Detection

As an effective feature learning method, deep learning has made breakthrough in applications such as computer vision, data mining, and speech recognition. In recent years, researchers have conducted in-depth research on pedestrian detection based on deep learning and have achieved abundant results. DBN-Isol (Isolated Deep Belief Network) [15] was proposed for part detection, which performs well on pedestrian detection in the presence of occlusion. ConvNet [19] consists of three convolutional layers, in which features obtained by the second and third layer are fused as input to the fully connected layer to finish pedestrian detection. DBN-Mut (Mutual Deep Belief Network) [20] extended the mutual visibility based on DBN-ISOL, which focuses on the situation where a pedestrian is partially blocked by another pedestrian during pedestrian detection. Unified Deep Networks (UDN) [21] was constructed

based on CNN (Convolutional Neural Network), part detection, a deformation model, and visibility reasoning. It makes full use of the advantages of DBN-ISOL and DBN-Mut, and it combines CNN and BP (Back Propagation) deep networks for pedestrian detection. SDN [22] combined feature learning, saliency mapping, and mixed feature representation in a cascade structure, in which a switchable RBM (Restricted Boltzmann Machine) layer is introduced on the traditional CNN to selectively combine different features. The above methods usually include two or three layers in the deep net.

To learn more sophisticate features, researchers implemented deeper networks to extract features, which usually include at least 16 layers. UDN was extend to a very deep net by using VGG16 and fast RCNN to obtain the features of different body parts for pedestrian detection [23]. RPN (Region Proposal Network) and faster RCNN (Regions with CNN features) were used simultaneously to detect and segment pedestrians in an image [24]; the backbone network is VGG16. PCN [25] uses an LSTM (Long Short-Term Memory) for part semantic learning, an RPN for region proposals, and a Maxout of different adaptive scale selection and the backbone network is also VGG16. By employing an attention network to the baseline faster RCNN, better detection was obtained [26]. Tesema et al. [27] extended the RPN (Region Proposal Network) + BF (Boosted Forest) framework [28] with handcrafted features and CNN features. ROI pooling is applied both on handcrafted features and CNN features for pedestrian candidate proposal. Extracted features are concatenated and sent to BF for classification. Brazil et al. [29] implemented a cascaded-phase design with a backbone network of VGG16 for progressive region proposal and pedestrian detection other than independent detection procedures with ROI systems. Zhang et al. [30] and Song et al. [31] employed ResNet with 50 layers for pedestrian feature extraction and obtained state-of-the-art results. Li et al. [32] utilized deep feature (feature maps from ResNet-50) pyramids for multi-resolution feature extraction and realized a competitive accuracy and real-time pedestrian detection on Geforce GTX GPU. Liu et al. [33] also utilized multi-resolution features from ResNet; the features are concatenated simultaneously for center point location regression and scale prediction without region proposal ahead.

However, in real scenarios of auto driving, the hardware computing ability is limited. Networks with too many layers are not practical, so a shallow network such as UDN is worth consideration.

## 3. Two-Stream UDN for Pedestrian Detection

For pedestrian detection, feature extraction, part deformation handling, occlusion handling, and classification are four important components. However, existing methods learn or design these components individually or sequentially. In order to maximize their strengths through cooperation, UDN [21] formulated the four components into a joint deep learning framework, which is a shallow network having two convolution layers.

### 3.1. Improved UDN

The UDN process images in YUV color space. The three channels fed to UDN include two images and one feature image, which is shown in the upper row of Figure 2. The first channel is the Y channel of an input image. The second channel consists of 4 small images and each is one-fourth of the original image; three of them are the Y, U and V channels of the original image, and the fourth part is blank image. The third channel also consists of 4 small images, and each is one-fourth of the original image, three of them are Sobel edge features computed with the Y, U, and V channel of the original image, the fourth part is achieved by maximizing the three edge images on pixel-wise. It can be seen that for some images, the UV channels may be too plain to provide enough information for recognition, and corresponding edge maps are nearly zeros, which is not benefit for feature extractions.

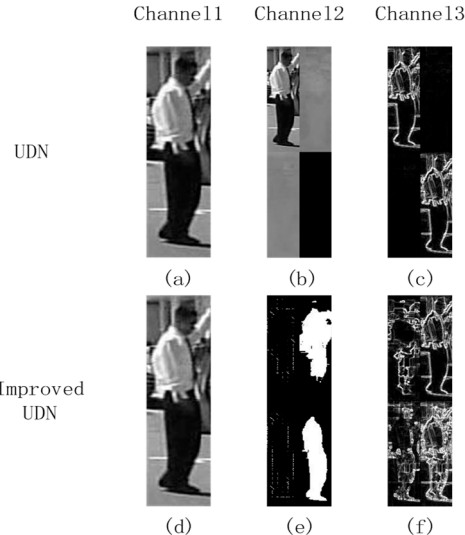

**Figure 2.** Input channels for Unified Deep Networks (UDN) and improved UDN. (**a**) The Y channel in YUV color space. (**b**) Concatenation of the Y channel, U channel, and V channel with zero padding. (**c**) Concatenation of four Sobel edge maps. The former three are obtained from three channel images in YUV color space, the fourth edge map is obtained by choosing the maximum magnitudes from the first three edge maps. (**d**) V channel in HSV color space. (**e**) Concatenation of two HOG feature maps and two Grabcut feature maps. (**f**) Concatenation of four Sobel edge maps. The former three are obtained from three channel image in HSV color space, the fourth edge map is obtained by choosing the maximum magnitudes from the first three edge maps.

In this paper, we employ HSV color space for feature extraction and provide more shape information for the network. The lower row of Figure 2 shows new input for networks. The three channels are as follows: (1) The first channel is the V channel extracted from the original image after HSV color space conversion; (2) The second channel is divided into four blocks, which are the HOG features of the HSV image, GrabCut regions of the HSV image, HOG features of the RGB image, and GrabCut regions of the RGB image. (3) The third channel consists of four blocks. The first three blocks are HSV edges, which are calculated by the $5 \times 5$ Sobel operator for the three channels of the HSV image, respectively. The fourth block is achieved by maximizing the three above edge images on pixel-wise. Compared with UDN, our proposed feature combination provides more shape information.

Figure 3 shows improved UDN, which has same structure with UDN but adopts different input features and different activation functions. The details are as follows:

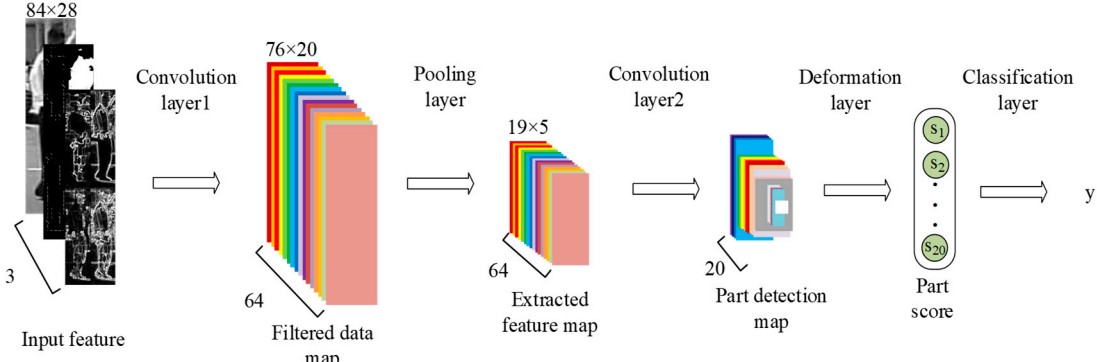

**Figure 3.** Improved UDN.

(1)   Convolution layer 1: The input of the first convolution layer has three components $X_i \in R^{84 \times 28}$, $i = 1, 2, 3$, which are corresponding to channels in Figure 2. There are 64 convolution

kernels $\{W_j | W_j \in R^{9 \times 9 \times 3}, \; j = 1, \ldots, 64\}$, which are utilized for extracting features. With each convolution kernel $W_j$, a filtered data map $A_j \in R^{76 \times 20}$ is computed as follows:

$$A_j \; = \; Swish\left(b_j \; + \; \sum_{i=1}^{3} [W_j]_{:,:,i} * X_i\right) \tag{1}$$

where $\left[W_j\right]_{:,:,i}$ represents the $i$th slice of three-order tensor $W_j$, * represents the convolution operator, and $b_j$ represents bias parameters obtained by random initialization. Before training, $W_j$ is initialized with the Gabor filter.

(2)　Pooling layer: Calculate the average value of pixels in each $4 \times 4$ neighborhood on each filtered data map and obtain the extracted feature maps $\{B_k | B_k \in R^{19 \times 5}, \; k = 1, \ldots, 64\}$, which is computed as follows:

$$B_k \; = \; AveragePool(A_k, 4, \, 4) \tag{2}$$

where $k$ represents the number of the extracted feature maps.

(3)　Convolution layer 2: The second convolution layer has 20 part-based filters $F_n \in R^{p_n \times q_n \times 64}$, $n = 1, \ldots, 20$ with different sizes. The filters are the same as with UDN. The part detection map $C_n$ is computed as follows:

$$C_n \; = \; Swish\left(u_n \; + \; \sum_{m=1}^{64} [F_n]_{:,:,m} * B_m\right) \tag{3}$$

where $n$ represents the number of the filtered data maps, $[F_n]_{:,:,m}$ represents the $m$th slice of the three-order tensor $F_n$, * represents the convolution operator, and $u_n$ represents the bias parameters obtained by random initialization. Before training, $F_n$ uses a Gabor filter for initialization.

(4)　Deformation layer: The deformation layer is same with the UDN [21] and returns score $s_P$ for the $p$th part, $\{s_P | p = 1, \ldots, 20\}$.

(5)　Classification layer: The classification layer is the same with the UDN [21] and estimates label y of the input image, indicating it is pedestrian or non-pedestrian.

The improved UDN adopts Swish [34] as activation function, which is different from Sigmoid used in UDN [21]. Otherwise, ReLU is the de facto standard activation function in deep learning in recent years, since it can solve the vanishing gradient problem, but its derivative is discontinuous. Swish is a smooth function. That means that it does not abruptly change direction as ReLU does near x = 0. Rather, it smoothly bends from 0 toward values <0 and then upwards again. Figure 4 shows the visualization of Sigmoid, ReLU, Softplus, Tanh, Swish, and FTSwish [35].

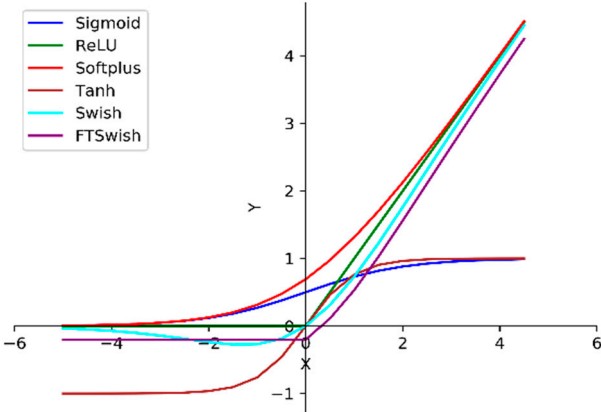

**Figure 4.** Common baseline activation functions. Best viewed in color.

Swish amends the drawback of Relu which zeros out negative values. When the training gradient decreases, Swish has a faster convergence rate than the traditional Sigmoid and other saturated nonlinear functions, which improves the training speed and guarantees the predictive performance of the network meanwhile. For an input $x$, its corresponding output transformed with Swish is $x(1 + e^{-x})^{-1}$.

### 3.2. Two-Stream UDN Based on Improved UDN

The key to pedestrian detection is to find the image area containing a human body. However, in an actual pedestrian detection scenario, some columnar objects such as trees and telephone poles are often misjudged as pedestrians, as shown in Figure 5.

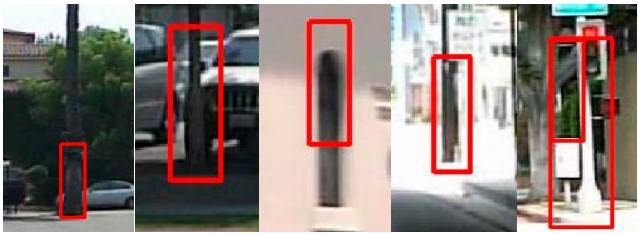

**Figure 5.** Examples of misjudged images with improved UDN.

Due to the relatively minor change of posture of the pedestrian's head, the region including the head has better invariance and discrimination from the upper part of columnar objects such as trees and telephone poles. Therefore, we construct another branch of improved UDN to accept the upper 1/3 pedestrian image as input.

This paper proposes a two-stream UDN network, which is shown in Figure 6. The two-stream UDN consists of two parallel branches: the Global network and the Local network. The Global network is an improved UDN described in Section 3.1, the details of each layer of the Local network are as follows:

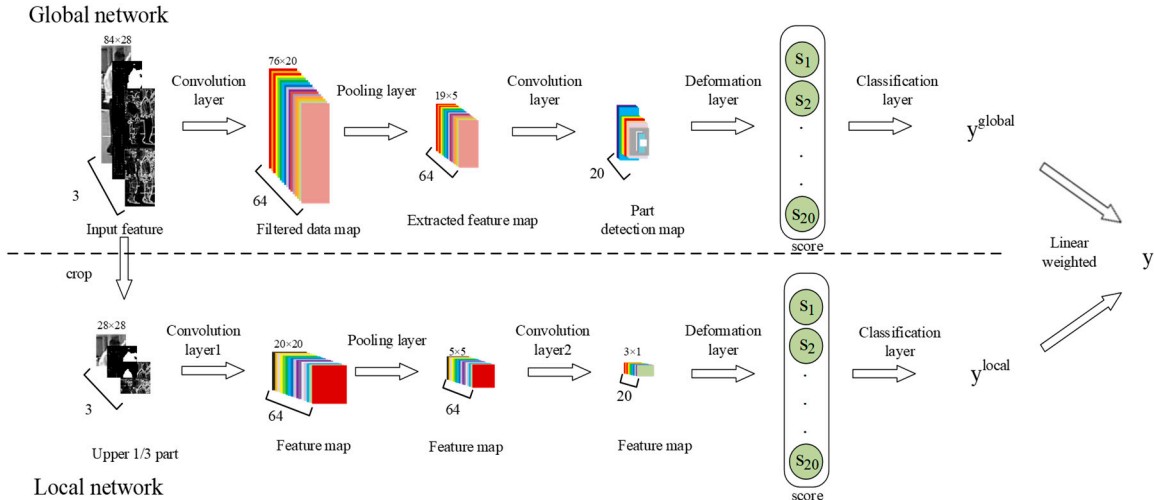

**Figure 6.** The two-stream UDN.

(1) Convolution layer 1: The input of the first convolution layer has three components $X_i^{local} \in R^{28 \times 28}$, $i = 1, 2, 3$, which are the upper 1/3 part region of the original input image corresponding to the lower input channels in Figure 6. There are 64 convolution

kernels $\left\{ W_j^{local} \middle| W_j^{local} \in R^{9\times9\times3}, \, j = 1, \ldots, 64 \right\}$ that are utilized for extracting features. With each convolution kernel $W_j^{local}$, the feature map $A_j^{local} \in R^{20\times20}$ is computed as follows:

$$A_j^{local} = Softplus\left( b_j^{local} + \sum_{i=1}^{3} [W_j^{local}]_{:,:,i} * X_i^{local} \right) \tag{4}$$

where $\left[ W_j^{local} \right]_{:,:,i}$ represents the *i*th slice of three-order tensor $W_j^{local}$, * represents the convolution operator, and $b_j^{local}$ represents the bias parameters for the local network, which is obtained by random initialization. Before training, $W_j^{local}$ is initialized with a Gabor filter.

(2) Pooling layer: Calculate the average value of pixels in each $4 \times 4$ neighborhood on each feature map $A_k^{local}$ and obtain extracted feature maps $\left\{ B_k^{local} \middle| B_k^{local} \in R^{5\times5}, \, k = 1, \ldots, 64 \right\}$, which is computed as follows:
$$B_k^{local} = AveragePool\left( A_k^{local}, 4, \, 4 \right) \tag{5}$$
where *k* represents the number of the extracted feature maps.

(3) Convolution layer 2: The second convolution layer has 20 filters $F_n^{local} \in R^{3\times5\times64}$, $n = 1, \ldots, 20$. Since the input of the local network is the upper 1/3 region of the image, there is no need to convolve in parts as in the network based on the global input. The local feature map of the second convolution layer $C_n^{local}$ is computed as follows:

$$C_n^{local} = Softplus\left( u_n^{local} + \sum_{m=1}^{64} [F_n^{local}]_{:,:,m} * B_m^{local} \right) \tag{6}$$

where *n* represents the number of the filtered data maps, $\left[ F_n^{local} \right]_{:,:,m}$ represents the *m*th slice of three-order tensor $F_n^{local}$, * represents the convolution operator, and $u_n^{local}$ represents the bias parameters obtained by random initialization. Before training, $F_n^{local}$ uses a Gabor filter for initialization.

(4) Deformation layer: The deformation layer is same with the global network, which reduces the number of parameters and extracts head–shoulder feature scores $s_p^{local}$, $\left\{ s_p^{local} \middle| p = 1, \ldots, 20 \right\}$.

(5) Classification layer: The visibility reasoning network [15] is used to estimate the label $y^{local}$ by learning the head–shoulder feature scores $s_p^{local}$, which represents whether the upper 1/3 part contains a pedestrian head–shoulder or not.

With the above Global network and the Local network, the two-stream UDN gives final prediction for input image with Equation (7) as the final output:

$$y = \beta y^{global} + (1 - \beta) y^{local} \tag{7}$$

where the value of $\beta$ is set by experience, and the details are shown in Section 4.2.1.

### 3.3. Training Tricks of Two-Stream UDN

The Global network and the Local network of the two-stream UDN network are first trained separately, and each subnetwork adopts the following variance loss shown in Equation (8):

$$J(W, b, x, y) = \frac{1}{m} \sum_{i=1}^{m} \left( \frac{1}{2} \| h_{W,b}(x^{(i)}) - y^{(i)} \|^2 \right) \tag{8}$$

where *y* represents the ground-truth label of the image, *x* represents the input of the network, *h* represents the prediction label of the input *x*, and *m* represents the number of total training samples.

After the two sub-networks are pre-trained independently, the Global network $N_G(W_1, b_1)$ and the Local network $N_L(W_2, b_2)$ are fixed temporally.

Then, we jointly train the two sub-networks with the loss function defined in Equation (9) to update the parameters of the global network and the local network, which are $(W_1, b_1)$ and $(W_2, b_2)$.

$$J(W_1, b_1, W_2, b_2; x, y) = \left[ \frac{1}{m} \sum_{i=1}^{m} \left( \frac{1}{2} \| \beta h_{W_1, b_1}(x^{(i)}) + (1 - \beta) h_{W_2, b_2}(x^{(i)}) - y^{(i)} \|^2 \right) \right] \tag{9}$$

During the joint training, the two sub-networks are updated alternately due to the differences in network input sizes and parameter volumes between the global and local networks.

Since pedestrian images are cut from street view videos captured by in-vehicle cameras, the number of positive samples is too small. In order to improve the generalization ability of the proposed method, it is necessary to expand the number of positive samples. This paper augments data by applying mirror transformation, clockwise rotation, and counterclockwise rotation to positive samples. The rotation angle is three degrees and bilinear interpolation is adopted during rotation. Figure 7 shows some examples of mirror transformation and angular rotation.

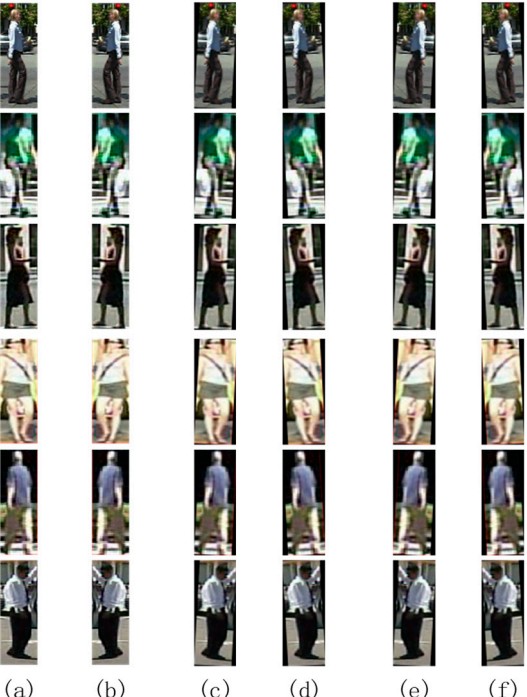

(a)　　(b)　　(c)　　(d)　　(e)　　(f)

**Figure 7.** The examples of positive sample augmentation. (**a**) Origin images; (**b**) images after a mirror flip; (**c**) images with a clockwise rotation of three degrees; (**d**) images with a counterclockwise rotation of three degrees; (**e**) mirrored images with a clockwise rotation of three degrees; (**f**) mirrored images with a counterclockwise rotation of three degrees.

## 4. Experimental Results

The training and testing images used in the experiments are same with Wanli Ouyang et al. [21], which are preprocessed from the Caltech pedestrian dataset using HOG + CSS + SVM. The original image size is $108 \times 36$. Before being fed into networks, images are cropped to $84 \times 28$. A positive sample is an image including a pedestrian, and a negative sample is an image without a pedestrian.

During network training, the input samples are randomly selected. A total of 60 samples are used in each batch, including 50 negative samples and 10 positive samples. The ratio of positive and negative samples is 1:5. The reason for choosing this ratio is to make it closer to what happens in real-life scenarios faced by the pedestrian detection system. With the augment method in Section 3.3,

approximately 60,000 negative samples and 12,000 positive samples are available for the INRIA pedestrian dataset [36], and approximately 60,000 negative samples and 24,000 positive samples are available for the Caltech pedestrian dataset [37].

The Log-average Miss Rate (LAMR) is used as a performance evaluation for evaluating different pedestrian detection methods, for which lower value means better performance.

In the process of parameter training, first we fix the learning rate to 0.05. When the LAMR does not decrease, we subtract 0.005 from the learning rate and continue until the learning rate is less than 0.02 and the LAMR no longer decreases. The network reaches an optimal solution.

### 4.1. Improved UDN

In this section, the improved UDN model is evaluated. The input images are preprocessed following the rules mentioned in Section 3.1.

#### 4.1.1. Comparison of Activation Functions

We experimented with Sigmoid, Relu, Softplus, Tanh, Swish, and FTSwish (T = −0.2) on a computer having Intel Core-i5 CPU. From Figure 8a, b, we can see that Swish outperforms other activation functions (AFs) with the quickest training loss decreasing and the quickest training accuracy increasing. Figure 8c represents the consuming time for running five epochs, and the y-axis represents times (seconds). Swish is also the least time-consuming AF. Therefore, using Swish as the activation function can significantly improve the calculation speed without sacrificing detection accuracy. In following experiments, the activation function we adopt is Swish.

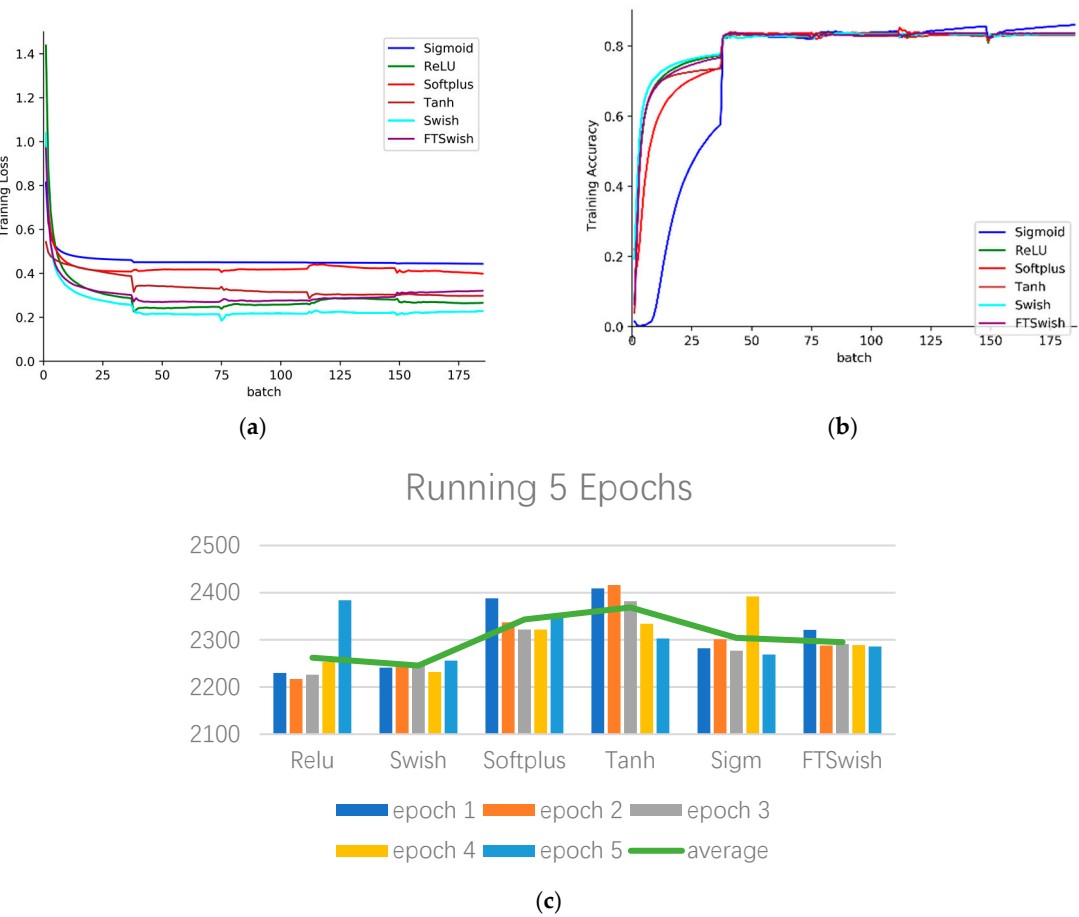

**Figure 8.** Comparisons of different activation functions (AFs). (**a**) Training loss; (**b**) Training accuracy; (**c**) Training time (in seconds) consuming for five epochs.

### 4.1.2. Comparison with Hand-Crafted Feature-Based Models

The first experiment trained the improved UDN with set00–set05 of the Caltech dataset, and it tested all models on set06–set10. The second experiment trained the improved UDN with the INRIA dataset and tested all models on the ETH dataset [38]. As seen in Table 1, the improved UDN is superior to the other models based on hand-crafted features, which obtains the lowest LAMR. Hand-crafted feature models such as VJ(Viola and Jones) [39], HOG [8], HOG+LBP(Local Binary Pattern) [40], ChnFtrs(Integral Channel Features) [10], ACF(Aggregated Channel Features) [41] obtain worse performance on both datasets. Figure 9 visualizes some pedestrian detection results, (a) shows images with ground truth with labeled pedestrians, and (b–g) are detection results of different models. False positive samples are in the red box, which means that negative samples are mistakenly predicted as pedestrians. True positive samples are in the green box, which means that pedestrians are correctly predicted. False negative samples are in the purple box, which means that these pedestrians are not detected by the methods. As we can see, the improved UDN obtains more green boxes and less red boxes and purple boxes. It is more effective than purely hand-crafted models.

**Table 1.** Log-average Miss Rate (LAMR) of improved UDN and hand-crafted feature-based models.

|         | VJ [39]  | HOG [8]  | HOG + LBP [40] | ChnFtrs [10] | ACF [41] | Improved UDN |
|---------|----------|----------|----------------|--------------|----------|--------------|
| Caltech | 94.73%   | 68.46%   | 67.77%         | 56.34%       | 51.04%   | 38.73%       |
| ETH     | 89.89%   | 64.23%   | 55.00%         | 57.00%       | 50.04%   | 45.04%       |

### 4.1.3. Analysis of Multiple Channel Inputs

Figure 10 shows the experimental results of investigating the influence of the different channel combinations introduced in Section 3.1. When the input data is the original three channels of RGB, the LAMR is 44.61%. When the input data is the first channel, the LAMR is 46.13%. When the input data includes both the first and the second channels, the LAMR is reduced by 4.77%. When the input data includes all three channels, the LAMR is reduced further by 2.63%.

The comparison results show that the original RGB image has the highest miss rate, which is lower than using the first channel alone. Combining more feature descriptions with the first channel, a much lower miss rate is obtained. Thus, the multi-channel feature input method can enhance the feature learning ability of the network and improve the recognition performance of the network.

### 4.2. Two-Stream UDN

In this section, experimental verification of the two-stream UDN for pedestrian detection is performed.

#### 4.2.1. $\beta$ Selection for Caltech and ETH

In Equation (7), $\beta$ represents the weights of the two subnetworks. We find the optimal value for the parameter $\beta$ on the Caltech and ETH dataset. Figure 11 shows the details. For both datasets, the Global network should be given more attention to reach a better LAMR, and the LAMR would rise if the Global network is focused on too much. For the Caltech dataset, the optimal value of $\beta$ is 0.78, and the according LAMR is 37.66%. For the ETH dataset, the optimal value of $\beta$ is 0.6, and the according LAMR is 44.58%. This shows that the global features are most important in pedestrian detection, and adding appropriate local features can improve the network's detection performance.

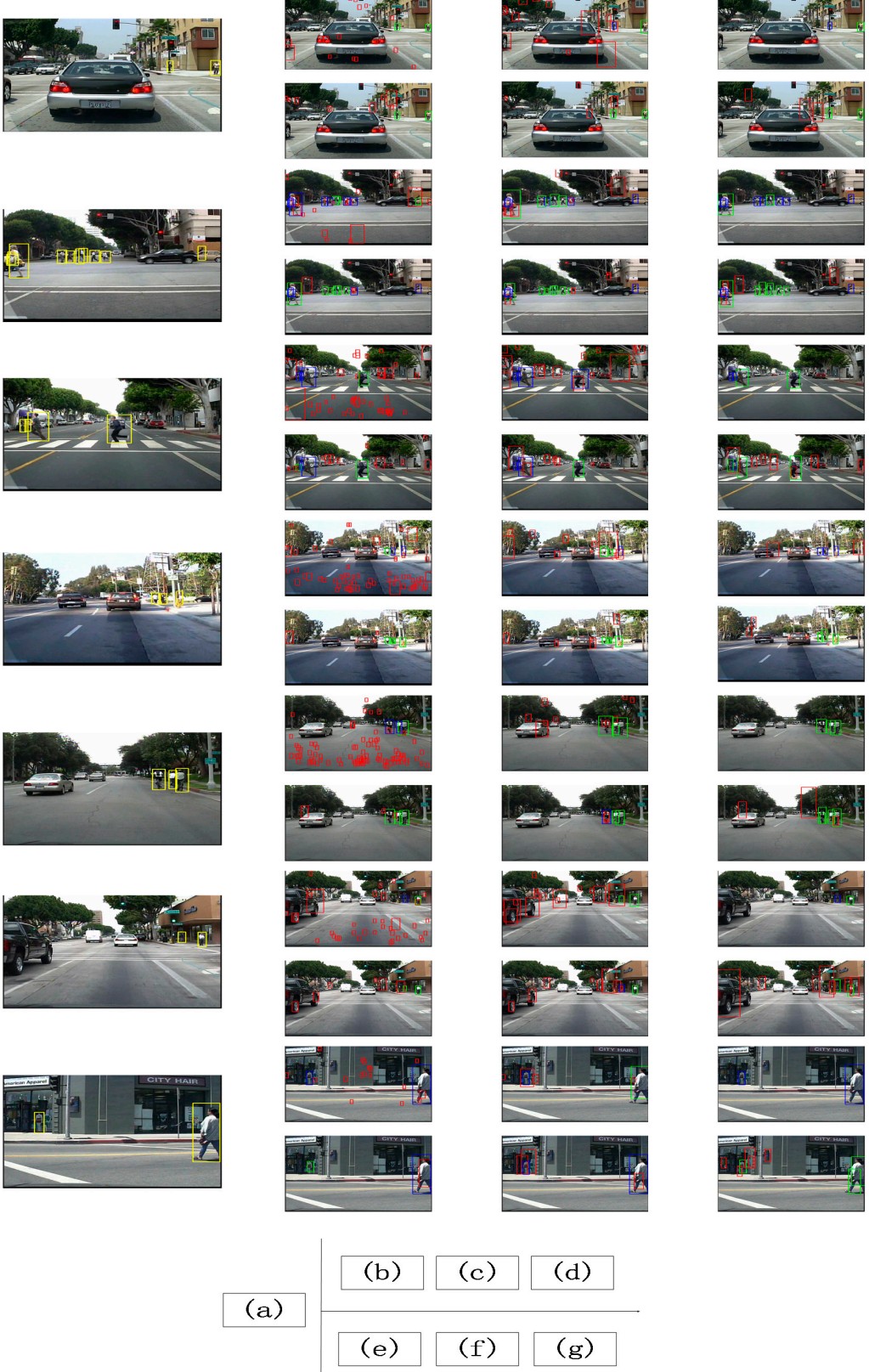

**Figure 9.** Detection results of models based on hand-crafted features and the improved UDN. False positive samples are in the red box. True positive samples are in the green box. False negative samples are in the purple box. (**a**) Ground Truth; (**b**) VJ; (**c**) HOG; (**d**) HOG + LBP; (**e**) ChnFtrs; (**f**) ACF; (**g**) Improved UDN. Best viewed in color.

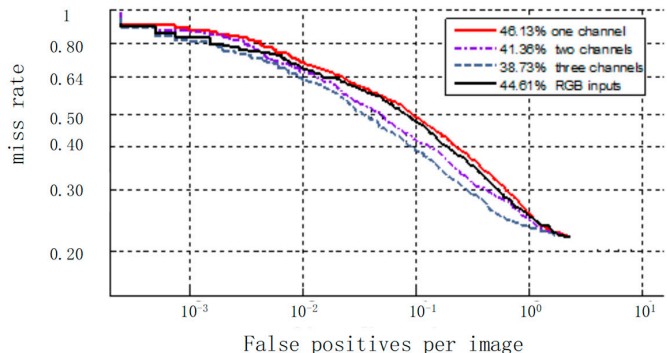

**Figure 10.** Results of various input channels with the Caltech test dataset.

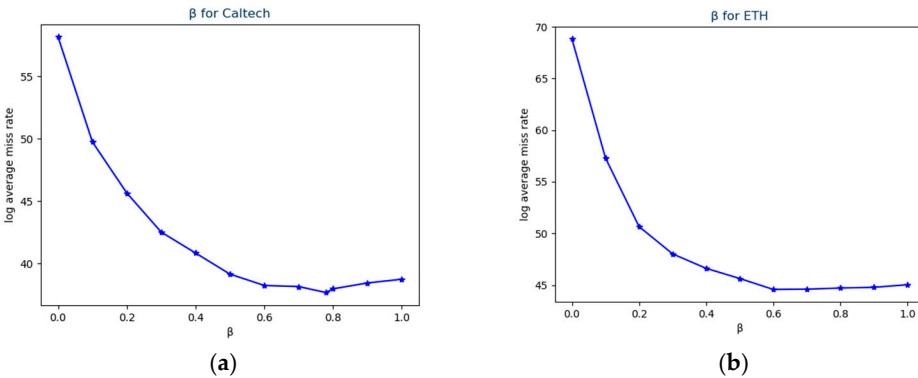

(**a**)    (**b**)

**Figure 11.** The effect of *β*. (**a**) Caltech test dataset; (**b**) ETH dataset.

### 4.2.2. Effectiveness of Adding Local Network

Table 2 shows the LAMR of the two-stream UDN and its two sub-networks on the Caltech and ETH datasets. As can be seen from Table 2, the LAMR of the two-stream UDN network for pedestrian detection on the Caltech dataset is 37.66%.

**Table 2.** LAMR of different networks.

|          | Global Network | Local Network | Two-Stream UDN |
|----------|----------------|---------------|----------------|
| Caltech  | 38.73%         | 58.15%        | 37.66%         |
| ETH      | 45.04%         | 68.82%        | 44.58%         |

Compared to the Global network, the two-stream UDN improved the miss rate by 1.07%. If only considering local features, the detection result is 58.15%, because the local network is only sensitive to a partial region, and it is easy to misjudge round objects as persons. Combing the Global network and Local network together, the results of the two-stream UDN network are better than each single network. A similar trend is shown on the ETH data set, which means joint training of global feature and local features are effective for pedestrian detection.

Scores of the two-stream UDN network and the Global network for true positive samples are shown in Figure 12. In Figure 12, (a) represents the score obtained by the Global network, and (b) represents the score obtained by the two-stream UDN. The results show that the Local network increased the confidence of the positive samples, which makes judgments for the true positive samples more accurate.

Scores of the two-stream UDN network and the Global network for false positive samples are shown in Figure 13. In Figure 13, (a) represents the core obtained by the Global feature, and (b) represents score obtained by the two-stream UDN. The aim of the local network is to find features that match the head and shoulders of pedestrians. Thus, negative samples whose upper 1/3 region

do not have head and shoulder features will obtain lower scores than that of the Global network. This illustrates that the extraction of head and shoulder features can be effectively enhanced by adding the Local network. Thus, the proposed two-stream UDN structure can improve the pedestrian detection results.

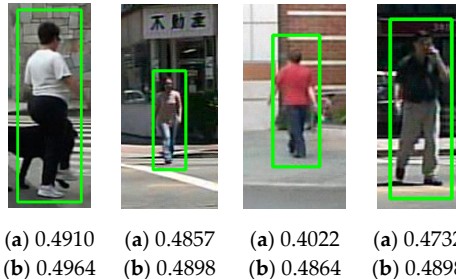

| (**a**) 0.4910 | (**a**) 0.4857 | (**a**) 0.4022 | (**a**) 0.4732 |
| (**b**) 0.4964 | (**b**) 0.4898 | (**b**) 0.4864 | (**b**) 0.4898 |

**Figure 12.** Scores for true positive samples (in green boxes) of the Caltech (**a**) scores by the Global network, (**b**) scores by the two-stream UDN.

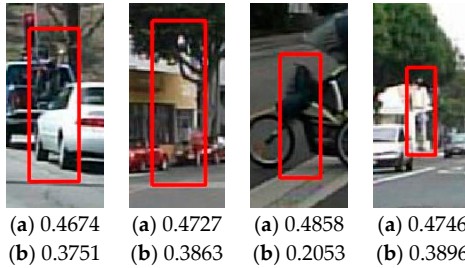

| (**a**) 0.4674 | (**a**) 0.4727 | (**a**) 0.4858 | (**a**) 0.4746 |
| (**b**) 0.3751 | (**b**) 0.3863 | (**b**) 0.2053 | (**b**) 0.3896 |

**Figure 13.** Scores for false positive samples (in red boxes) of the Caltech. (**a**) Scores by the Global network, (**b**) scores by the two-stream UDN.

### 4.2.3. Comparison with Other Shallow Nets

Table 3 shows comparison of the two-stream UDN and other shallow nets on Caltech and ETH. All the compared networks have two or three layers, and all their input features are hand-crafted features, such as HOG, gradient, or color self-similarity (CSS).

**Table 3.** LAMR of two-stream UDN and shallow networks.

|         | DBN-Isol [15] | DBN-Mut [20] | UDN [21] | SDN [22] | MultiSDP [42] | Two-Stream UDN |
|---------|---------------|--------------|----------|----------|---------------|----------------|
| Caltech | 53.29%        | 48.22%       | 39.32%   | 37.87%   | 45%           | 37.66%         |
| ETH     | 47.01%        | 41.07%       | 45.32%   | 40.63%   | 48%           | 44.58%         |

Comparing with other shallow nets, the improved UDN gets the lowest score on the Caltech and the second lowest score on the ETH. Comparing with the UDN, the improved UDN improves more obviously on the Caltech. A possible reason is that images in ETH and INRIA have higher resolution, so the global network can extract better features on the upper region, so the local network benefits less. The results indicate that the two-stream UDN works better when the captured image has low resolution. Figure 14 shows some samples in which the two-stream UDN outperforms the other models. False positive samples are in the red box, which means that the negative samples are mistakenly predicted as pedestrians. True positive samples are in the green box, which means that pedestrians are correctly predicted. False negative samples are in the purple box, which means these pedestrians are not detected by the methods. Compared with UDN [21], the two-stream UDN obtains fewer red boxes, which benefits from the Local network for eliminating non-pedestrian samples. Compared with SDN (Switchable Deep Network) [22], which has a similar LAMR with the two-stream UDN, the two-stream UDN predicts more green boxes (Figure 14, image 1, 2, 5 and 7). SDN is a network that

also emphasizes local features, so it receives a similar score with the proposed two-stream UDN. The SDN gets low-level local feature maps directly divided from low-level global feature maps, but the two-stream UDN learns the local feature maps with a separate branch.

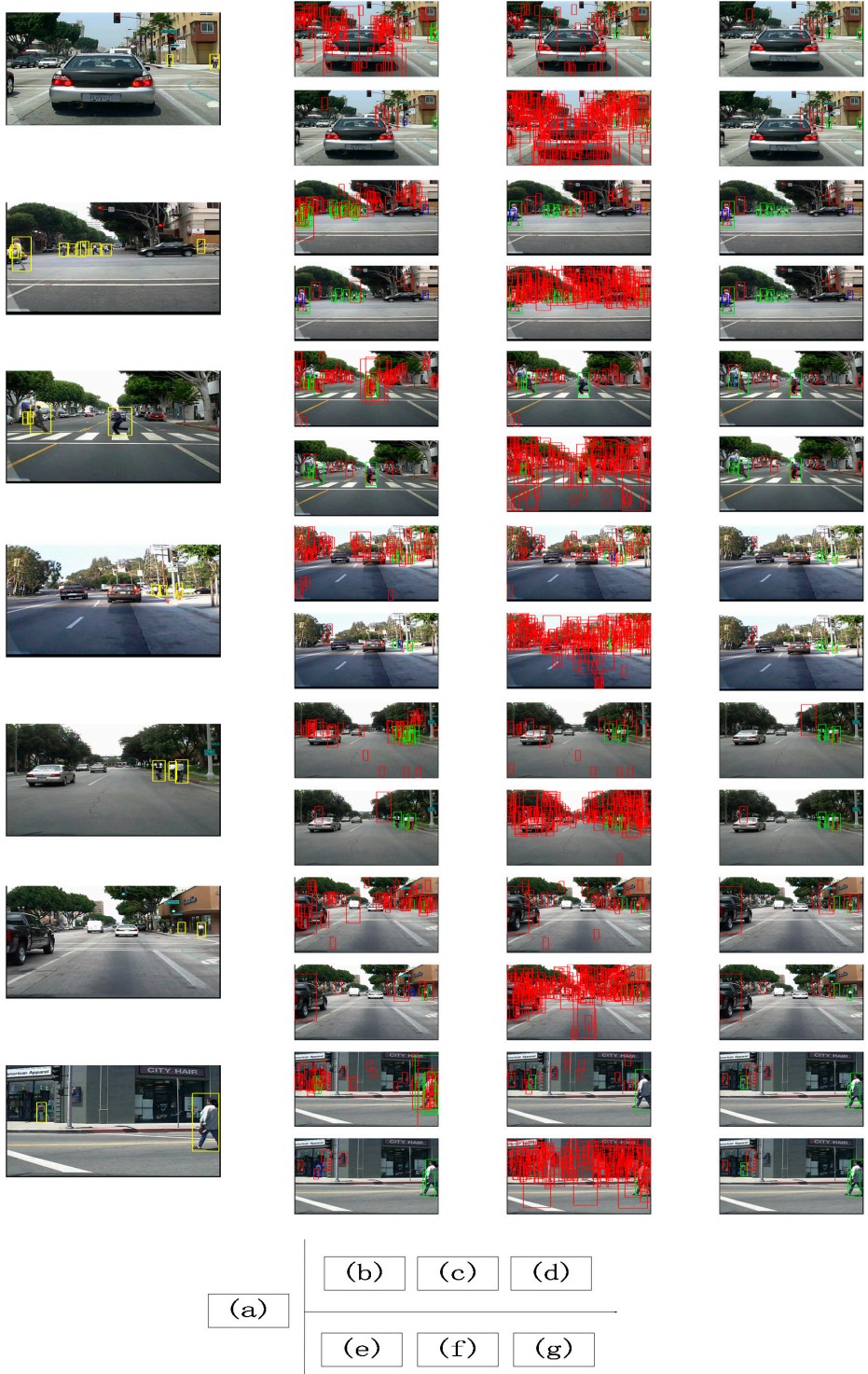

**Figure 14.** Detection results of models based on shallow networks. False positive samples are in the red box. True positive samples are in the green box. False negative samples are in the purple box. (**a**) Ground truth; (**b**) DBN-Isol; (**c**) DBN-Mul; (**d**) UDN; (**e**) SDN; (**f**) MultiSDP; (**g**) Two-stream UDN (ours). Best viewed in color.

### 4.2.4. Computational Complexity Analysis

For comparing the computation efficiency of different models, we have computed the model scale and evaluated the running time, which are shown in Table 4. Table 4 also shows the corresponding hardware for achieving the running time. All the deep networks were executed on computers with GPU, the shallow net SDN is also executed on computer with GPU. UDN and the proposed two-stream UDN are executed on the computer with CPU. Although executed on the computer without GPU, the proposed method is much faster than SDN [22] executed on the computer with GPU, which has a competitive detection miss rate with our model. Very deep models such as CSP (Center and Scale Prediction) [33] back boned with ResNet-50 run on GPU obtain approximately the same speed with our model run on CPU and have much larger model parameters. Even the backbone network is changed to squeezed models such as MobileNetV1; their model is still very large and the speed-up execution time highly relies on expensive GPU hardware.

**Table 4.** Comparisons of running time among the state-of-the-art methods.

| Model Type | Method | Hardware | TOPS | Time/img (s) | Parameters |
|---|---|---|---|---|---|
| | SDN [22] | GTX 760 GPU | 76 | 0.1 | 1M+ |
| Shallow Nets | UDN [21] | Intel Core i5 CPU | 0.46 | 0.051 | 53k |
| | Two-stream UDN (ours) | Intel Core i5 CPU | 0.46 | 0.076 | 89k |
| | CCF [43] | Titan Z GPU | 968 | 13 | 30k |
| | CSP (MobileNetV1) [33] | GTX 1080Ti GPU | 352 | 0.041 | 12.6M |
| Deep Nets | CSP (ResNet-50) [33] | GTX 1080Ti GPU | 352 | 0.061 | 40.6M |
| | Feature Pyramid [32] | GTX 1070 GPU | 206 | 0.13 | 64M |
| | RPN + BF [28] | Tesla K40 GPU | 243 | 0.5 | 230M |

Compared with the baseline model improved UDN (which has the same model size with UDN), our two-stream UDN model adds a subnet (Local network), which brings approximately 46,000 parameters increase and 0.025 s running time per image. Although the improvement of LAMR is only 1.07%, the Local network mainly reduces false positive samples, which is what the subnet designed for. As for the detection results of Caltech, the two-stream UDN reduces false positives by 8.6% compared with the improved UDN.

### 5. Conclusions

This paper proposed a two-stream UDN for pedestrian detection, which includes two sub-networks trained jointly and used for judgement independently. The two sub-networks give prediction based on the overall or the upper 1/3 region of the bounding box, and they accept the V channel in HSV space, HOG feature map, Sobel feature map, and GrabCut feature map as input.

In recent years, many deep learning methods [28,32,33] have obtained excellent results by extracting and combining complex deep features for pedestrian detection, but the models based on deep networks have critical requirements for the hardware shown in Table 4, and TOPS (Tera Operations Per Second) are computed according to the specifications [44–49]. While the pedestrian detection is applied in an auto-driving platform, it cannot obtain huge computation resources. Table 5 lists the usual computation resources of popular auto-driving platforms. The proposed two-stream UDN can execute on a computer CPU with the speed of 0.076 s/img, which indicates that it can run on the hardware resources of the autonomous driving platform. Except for the reducing requirement for the hardware, improving true positive detections and reducing false positive detections are also important to promote the application of pedestrian detection, considering the environment context during the decision process is feasible, such as parameters whereby a pedestrian should not be located in the sky or on the surface of a car.

**Table 5.** Main product performance of self-driving chips.

| Product Name | Company | TOPS | TTM |
|---|---|---|---|
| Journey [50] | Horizon | 4 | 2017 |
| EyeQ4 [51] | Mobileye | 2.5 | 2018 |
| Drive Xavier [52] | NVIDIA | 32 | 2018 |
| EyeQ5 [51] | Mobileye | 24 | 2020 |

**Author Contributions:** Conceptualization, W.W. and L.W.; methodology, L.W.; software, W.W. and X.G.; validation, W.W., L.W. and X.G.; formal analysis, W.W.; investigation, X.G.; resources, B.Y.; data curation, X.G.; writing—original draft preparation, W.W.; writing—review and editing, W.W., L.W. and J.L.; supervision, B.Y.; project administration, B.Y.; funding acquisition, B.Y. All authors have read and agreed to the published version of the manuscript.

**Funding:** This work is funded by National Natural Science Foundation of China (61876012, 61772049), and Beijing Natural Science Foundation (4202003).

**Conflicts of Interest:** The authors declare there is no conflicts of interest regarding the publication of this paper.

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
