# Peer review of "Pedestrian Detection Based on Two-Stream UDN"

_applsci, doi:10.3390/app10051866_

Round 1

Reviewer 1 Report

The paper presents a two-stream UDN for pedestrian detection for systems with low computational power. The system is well presented, but the Result section could be improved with some more figures.

In particular:

In equation 7, how do you select the beta values? You say that it "is set by experience." What does it mean? Is it trained or selected by the user? IN the second case, what value do you use? And is this value always the same for all the datasets, or it changes? Please explain better. In sections 4.1.2 and 4.2.2 you should add some figures that show the case where the other networks fail and your is the best one. In figure 9, what is the meaning of the double red rectangular shape? In figure 10, are only false samples? It seems that in some images, there are some pedestrians, as the third one.  It would help if you discussed better why the SND[22] has a score very similar to the proposed two-stream UDN.

Reviewer 2 Report

The paper describes a pedestrian detection method called two-stream UDN. The proposed approach is a modified version of the UDN model from 2013. Main contributions are: (1) a modification of the original UDN model by changing the input data and the activation function; (2) an addition of the second path in the model consisting of the smaller network that operates on the upper third of input images. Overall, the paper has a clear structure, however related work section could be expanded to include most recent state-of-the-art works on pedestrian detection, for instance, [1-4]. Introduction gives a good motivation for solving pedestrian detection and Related Works gives a good overview of different approaches.

My biggest concern here is that the authors do not clearly formulate a specific gap in the current knowledge that they’re trying to address and do not explain how is’t going to be done in the rest of the manuscript. I suggest to state clearly:

  • what is current state of using deep networks for pedestrian detection and what are the critical disadvantages existing methods have that this paper is addressing
  • clearly state the goal of the study and define the way to validate that the proposed method in fact addresses issues defined above
  • state the practical implications for the field that will come from using this new method

For example, so far for deep learning-based pedestrian detection, the authors only identified one issue, that “in real scenarios of auto-driving, the hardware computing ability is limited. Networks with too much layers are not practical”. If that is the main problem that the paper is trying to address, I find this formulation too vague and not specific. There are many approaches to low-resource deep learning for object detection (mobile nets, quantization, etc., see [5]) and they cannot be ignored, if the limited compute is the main issue here. To give an example, lightweight VGG-type network running on FPGA is capable of real-time video recognition [6]. However, the authors do not define the type of “limited hardware” they’re considering in this case. Even more unfortunate, that there is no discussion of this problem further in the paper, except for when the authors choose other models to compare their method with. When performance is important, one could expect some comparison in terms of model size in parameters, memory footprint, and speed, e.g. in FPS. For examples of studies performing such analyses, please refer to [1] and [3]. 

Methodology section is mostly technically sound, although the proposed approach is based on the dated method for which many improvements have been suggested since it came out in 2013. For the same reason, the authors should clarify the choice of activation functions. First, they do not compare ReLU with Softplus, although ReLU is mentioned in methods. Then, there have been many AFs suggested since the original UDN paper came out in 2013, e.g. see [7]. If changing AF is one of the main methodological contributions, then comparing with other more recently proposed AFs would strengthen the claim.

Evaluation of another contribution–Local network–also misses quantification of processing costs that come with the second path addition. Since the performance improvement is just over 1%, it would be helpful to measure how much this modification affects model weight and processing speed.

The Conclusion should discuss the meaning of the results for the broader context. It should loop back again to the main goal of the paper that is currently missing from the Introduction.

References:

[1] Li, Zhaoqing, et al. "Real-time pedestrian detection with deep supervision in the wild." Signal, Image and Video Processing 13.4 (2019): 761-769.

[2] Brazil, Garrick, and Xiaoming Liu. "Pedestrian detection with autoregressive network phases." Proceedings of the IEEE Conference on Computer Vision and Pattern Recognition. 2019.

[3] Liu, Wei, et al. "High-level semantic feature detection: A new perspective for pedestrian detection." Proceedings of the IEEE Conference on Computer Vision and Pattern Recognition. 2019.

[4] Tesema, Fiseha B., et al. "Hybrid Channel Based Pedestrian Detection." Neurocomputing (2020).

[5] Cheng, Yu, et al. "A survey of model compression and acceleration for deep neural networks." arXiv preprint arXiv:1710.09282 (2017).

[6] Solovyev, Roman, et al. "Fixed-Point Convolutional Neural Network for Real-Time Video Processing in FPGA." 2019 IEEE Conference of Russian Young Researchers in Electrical and Electronic Engineering (EIConRus). IEEE, 2019.

[7] What is the FTSwish activation function? https://www.machinecurve.com/index.php/2020/01/03/what-is-the-ftswish-activation-function/

Reviewer 3 Report

Conclusions presented in the article are very general and dosen't contain any suggestions about further work or area of development. It would be helpful for other scientist to have such information and shows that you analyzed problem and your solution with multiple points of view. Also I'd suggest to improve the discussion about the results. After the lecture, I'm under impression that you presented set of numbers in both table and written way but there could be more comments about their meaning.
